# Non-contact detection of pyrethroids widely used in vector control by *Anopheles* mosquitoes

Sassan Simplice Kambou[1,2]*, Adeline Valente[1], Philip Agnew[1], Domonbabele François de Sales Hien[2], Rakiswendé Serge Yerbanga[1,3], Nicolas Moiroux[1], Kounbobr Roch Dabire[2], Cédric Pennetier[1], Anna Cohuet[1], David Carrasco[1]

1 MIVEGEC, University Montpellier, IRD, CNRS, Montpellier, France, 2 Institut de Recherche en Sciences de la Santé (IRSS), Centre National de Recherche Scientifique et Technique (CNRST), Bobo-Dioulasso, Burkina Faso, 3 Institut des Sciences et Techniques (InSTech), Bobo-Dioulasso, Burkina Faso

* simplice.kambou@ird.fr

**Data Availability Statement:** The data and related documentations that support the findings of this study are openly available in DataSuds repository (IRD, France) at https://doi.org/10.23708/IMJ1WG.

## Abstract

Pyrethroids are the most widely used insecticides to control vector borne diseases including malaria. Physiological resistance mechanisms to these insecticides have been well described, whereas those for behavioral resistance remain overlooked. Field data suggest the presence of spatial sensory detection by *Anopheles* mosquitoes of the pyrethroid molecules used in insecticide-based control tools, such as long-lasting insecticide nets or insecticide residual spraying. This opens the way to the emergence of a wide range of behavioral adaptations among malaria vectors. However, the spatial sensory detection of these molecules is controversial and needs to be demonstrated. The goal of this study was to behaviorally characterize the non-contact detection of three of the most common pyrethroids used for malaria vector control: permethrin, deltamethrin an α-cypermethrin. To reach this goal, we recorded the behavior (takeoff response) of *Anopheles gambiae* pyrethroid-sensitive and resistant laboratory strains, as well as field collected mosquitoes from the Gambiae Complex, when exposed to the headspace of bottles containing different doses of the insecticides at 25 and 35°C, in order to represent a range of laboratory and field temperatures. We found the proportion of laboratory susceptible and resistant female mosquitoes that took off was, in all treatments, dose and the temperature dependent. Sensitive mosquitoes were significantly more prone to take off only in the presence of α-cypermethrin, whereas sensitive and resistant mosquitoes showed similar responses to permethrin and deltamethrin. Field-collected mosquitoes of the Gambiae Complex were also responsive to permethrin, independently of the species identity (*An. gambiae*, *An. coluzzii* and *An. arabiensis*) or their genotypes for the *kdr* mutation, known to confer resistance to pyrethroids. The observed ability of *Anopheles* spp. mosquitoes to detect insecticides without contact could favor the evolution of behavioral modifications that may allow them to avoid or reduce the adverse effect of insecticides and thus, the development of behavioral resistance.

Data and documentations reuse is granted under CC-BY license. R script is granted under GPLV3 license.

**Funding:** This study was financed by the French National Research Agency – ANR (https://anr.fr), project INDEed (ANR-21-CE35-0021-01) granted to DC. The "Institut de Recherche pour le Développement" through the program ARTS provided funding for SSK doctoral studies. the funders had no role in study design, data collection and analysis, decision to publish, or preparation of the manuscript.

**Competing interests:** The authors have declared that no competing interests exist.

## Introduction

Insecticide-based vector control remains the main component for the prevention of malaria transmission. Since the beginning of the century, large scale implementation of insecticide treated nets (ITNs) and indoor residual spraying (IRS) has resulted in a significant reduction of malaria cases [1]. Pyrethroids are the most frequently and widely used class of insecticides to control malaria vector species (*Anopheles spp.*), due to their low toxicity shown in mammals [2]. However, the extensive use of pyrethroids to control pest insects, including in agricultural practices, has favored the selection and spread of insecticide resistance within malaria mosquito populations [3–8]. Thanks to the continuous efforts made in the last decades, researchers have been deciphering the different mechanisms behind physiological insecticide resistance in mosquitoes, such as target site resistance [4,9,10], metabolic resistance [4,11] or cuticular resistance [4,12,13]. Besides physiological resistance mechanisms, the emergence of behavioral resistance–i.e., behavioral adaptations that help mosquitoes to partially or completely overcome the deleterious effects of insecticides–have been recognized and may also play an important role as a mechanism reducing the efficacy of insecticide-based vector control tools [14,15]. While less documented than physiological resistance, several studies evidenced behavioral modifications in *Anopheles* species subsequent to the implementation of insecticide-based control methods in the field, as for instance: increased mosquito outdoor host-seeking (i.e. spatial avoidance) [16–18], host shifts (i.e. trophic avoidance) [18,19] or biting daily-rhythm shifts (i.e. temporal avoidance) [20–22]. Such behavioral modifications may result from the evolution of either constitutive behavioral resistance traits (i.e., selection of genetic behavioral variants over generations) or inducible behavioral resistance traits (i.e., phenotypic plasticity within a generation) [14]. The latter implies that mosquitoes are able to detect the insecticides through their sensory system and therefore modify their behavior accordingly.

Proof of non-contact pyrethroid sensory detection has been deduced from the avoidance response mosquitoes display against certain pyrethroid molecules, particularly the so called "volatile/volatilized pyrethroids" commonly used as spatial repellents: e.g., metofluthrin, trans-fluthrin or prallethrin [23–26]. However, it is less clear whether other pyrethroids commonly used in IRS and ITNs, such as permethrin, deltamethrin or alpha-cypermethrin, may elicit a behavioral response in mosquitoes before contact (i.e., from a distance). According to their vapor pressure values at 25°C: e.g., permethrin $= 1.48 \times 10^{-8}$ mm Hg, deltamethrin $= 9.32 \times 10^{-11}$ mm Hg and cypermethrin $= 2.5 \times 10^{-9}$ mm Hg [27], these pyrethroids are considered as semi- or non-volatile molecules, and such low-volatility is not in favor of their remote detection by mosquitoes [28,29]. Nevertheless, insecticides applied for vector control in malaria endemic regions are exposed to higher temperatures, e.g., 35°C-39°C [30–32]. Such temperatures can increase the vapor pressure values of these compounds and thus their volatility, as predicted by the Clausius-Clapeyron equation. Field studies have provided indirect evidence for the potential spatial detection of these chemical compounds in mosquitoes. For instance, meta-analysis studies showed that in experimental huts, malaria vector species (*Anopheles spp.*) movement (i.e., entrance and exiting) was affected by the presence of ITNs in the huts [33,34]. Depending on the experimental treatments, the results unveiled both: a deterrent and an attractive effect, related to the presence of ITNs in the huts, suggesting that malaria vectors might be able to detect ITNs from a distance before entering the hut. Another study using a dual-choice olfactometer assay found that knock-down resistant (*kdr*) homozygous *An. gambiae* mosquitoes, thus resistant to pyrethroids, were more attracted by a host placed behind an ITN than an untreated net, while the presence of insecticide on the net did not affect the choice of susceptible mosquitoes [35]. In other mosquito genera, a recent study evidenced an avoidance response of *Culex quinquefasciatus* and *Aedes aegypti* to permethrin, deltamethrin and λ-cyhalothrin, but

only when individuals had been previously exposed (by contact) to the molecules before the test [36]. Overall, these results showing either an attraction or an avoidance of pyrethroid compounds by mosquitoes implies the detection of these molecules from distance. Conversely, other studies investigating *Anopheles* mosquitoes-insecticide interactions did not find any sign of remote repellency in mosquitoes prior to contact [37–42] and the characterization of the distance for the detection of insecticides used in vector control by the targeted mosquitoes remains to clarified.

This study aims to determine the non-contact detection of low volatile compounds (permethrin, deltamethrin and alpha-cypermethrin) in laboratory-reared susceptible and *kdr* resistant *An. gambiae* s.s. at two different temperatures, 25˚C and 35˚C, that represent a range of realistic conditions. In addition, in order to generalize the results, we tested non-contact detection in wild caught *An. gambiae*, *An.coluzzii* and *An.arabiensis* mosquitoes.

## Materials and methods

### Mosquito rearing

Two reference strains of *An. gambiae sensu stricto* were used in this study. The first is the insecticide-susceptible reference strain Kisumu (hereafter called "kis" strain), which originates from a field collection in Kenya in 1953 [43]. The second is the pyrethroid resistant strain "kdrkis", homozygous for the *kdr* mutation. This strain was obtained by introgression of the *kdr*-west allele, harboring the L1014F mutation in the voltage-gated sodium channel gene into the Kisumu genome. The *kdr*-west allele was obtained from pyrethroid resistant mosquitoes sampled in Valley du Kou, Burkina Faso [44]. Both colonies were reared at the "Vectopôle" insectary at the "Institut de Recherche pour le Développement (IRD)" in Montpellier (France) and at the "Institut de Recherche en Sciences de la Santé (IRSS)" in Bobo-Dioulasso (Burkina Faso). Adult mosquitoes were maintained under standard insectary conditions (fed with 10% of honey solution, maintained at 27 ±1˚C and 70–80% relative humidity and under a 12h:12h light:dark cycle). *An. gambiae sensu lato* L1 and L2 stage larvae were collected in the field at Soumousso (11˚ 00′ 41″ N, 4˚ 02′ 50″ W) in Burkina Faso. Larvae were transferred into tubes for transportation to the insectary at the IRSS and placed under the same conditions as the laboratory strains. Adult females (fed with sucrose 10%) aged from 4 to 8 days post-emergence were used for the behavioral assays. This choice of age was made to assure a high proportion of mated females, consequently reducing the potential variability due to insemination status [45]. Species identity was determined for each female by PCR [46] after behavioral tests. Genotyping for *kdr* mutations known to confer resistance to pyrethroids (*kdr* West and *kdr* East) was carried out [44,47] as well as genotyping for Ace-1(R) mutation [48] known to confer resistance to carbamates and organophosphates.

### Chemical solutions used

Permethrin (Sigma-Aldrich, CAS #: 52645-53-1, product #: 45614, batch #: BCCF2182, purity: 95%), deltamethrin (Sigma-Aldrich, product #: 45423, batch #: BCCF1130, CAS #: 52918-63-5, purity: 98.6%) and alpha-cypermethrin (Sigma-Aldrich, product #: 45806, batch #: BCCF16154, CAS #: 67375-30-8, purity: 98.2%) stock solutions were prepared at 1mg/mL in acetone (Sigma-Aldrich, Product #: 650501, CAS #: 67-64-1, purity: 99.99%). In this study, the dose corresponds to the mass of insecticide in micrograms (μg) placed into the test bottle during the experiment, as we let the acetone completely evaporate under a chemical hood from 5 min to 30 min depending on the dose. Four or five doses of pyrethroids were tested during the experiments: 4μg, 40μg, 400μg, 800μg (not used for permethrin) and 1600μg. Acetone was

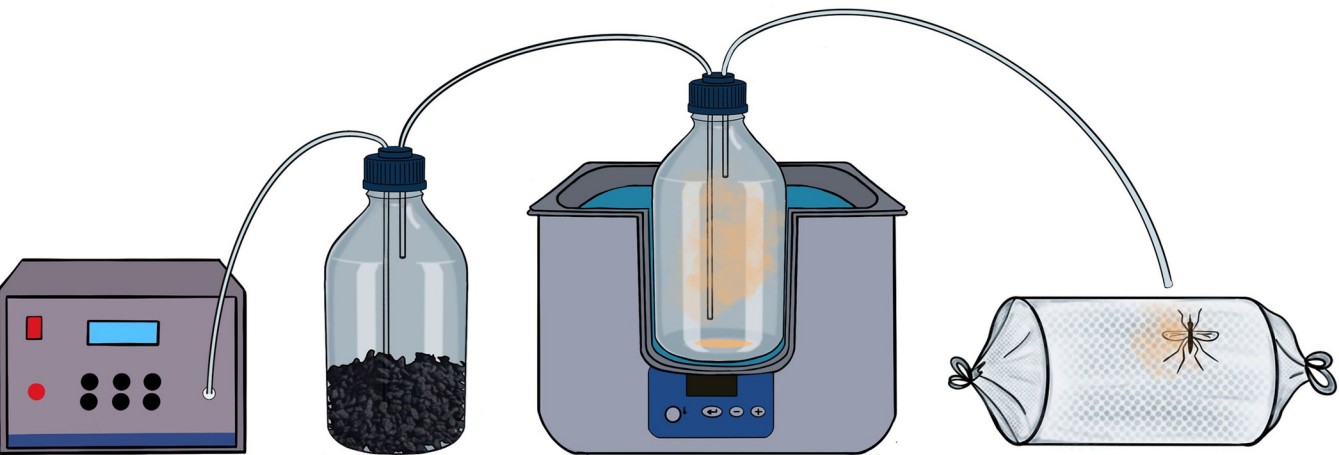

**Fig 1.** Schematic set-up of the behavioral assay (a: Stimulus controller; b: Gas washing bottle containing activated charcoal; c: Water bath; d: Sample bottle for insecticide or control solution; e: PTFE tubing; f: Cage containing one mosquito female).

used as a negative control in the behavioral assays and it was treated in a similar manner to the insecticide samples.

## Behavioral assay set-up description

In this behavioral assay inspired from repellent detection assays [49], we determined whether adult female *An. gambiae s.l.* detected permethrin, deltamethrin and alpha-cypermethrin without contact based on their behavioral response (i.e., takeoff). Females were exposed to the headspace of varying doses of insecticides disposed at two different temperatures: 25°C and 35°C. A set-up including a sample bottle containing the dose of pyrethroid to be tested was placed into a water bath to adjust the headspace temperature to either 25°C or 35°C (Fig 1). Headspace temperature was measured using a digital thermometer (model t110, Testo AG, Germany). A 30s continuous charcoal-purified airflow (0.48L/min) at 0.16 cm$^2$/s was circulated by means of a stimulus controller (CS-55, Syntech, Germany), through the sample bottle and directed by polytetrafluoroethylene (PTFE) tubing towards a single female mosquito confined in a cylindrical cage (10 × 20 cm) enclosed by mosquito netting. The end of the tube was manually placed (by an operator wearing laboratory coat and nitrile gloves) at approximately 1–2 cm from the female mosquito. Mosquitoes from either a laboratory colony or field collected individuals were tested individually at room temperature (25–27±1°C).

A mosquito's behavioral response was recorded as a binary variable (i.e., takeoff: yes/no) during the 30s maximum period of exposure to the airflow. For each replicate, 10 individuals were tested per treatment (insecticide *versus* control at both temperatures). Three to five replicates were carried out depending on the mosquitoes involved (i.e., the two laboratory strains and field mosquitoes). For project related reasons, assays involving permethrin on laboratory strains were carried out in Montpellier, France. Assays involving deltamethrin and alpha-cypermethrin on laboratory strains were carried out in Bobo Dioulasso, Burkina Faso. Field collected mosquitoes were only tested with permethrin in Bobo Dioulasso, Burkina Faso.

## Statistical analysis

All analyses were performed in R v.4.3.1. [50]. Initial analyses found that the proportion of females taking off in the control groups differed according to the insecticide test involved: 12.6% of mosquitoes in the control group took off during the assays testing the effect of

permethrin, 29.6% in the control group during the assays for deltamethrin, and 32.5% in the control group during the assays for alpha-cypermethrin (see S1a and S1b Fig in S1 File). These differences were temporally and physically confounded with the locations in which tests were made. Consequently, the data associated with each insecticide were analyzed separately.

Within the dataset for each insecticide, analyses found the proportion of females responding in the control treatments varied little among treatments or replicates and did not depend on the strain, temperature or dose treatments involved (see S2a-S2c Fig in S1 File). Consequently, the control data within each insecticide dataset were pooled together to provide a single estimate for the proportion of mosquitoes responding in the control treatments of each dataset.

The proportion of mosquitoes responding to the control treatments in each dataset was used as a baseline against which the proportions of mosquitoes responding in the insecticide stimulus treatments were compared (see S2d-S2f Fig in S1 File). The ratio of the latter to the former estimates the relative risk of whether mosquitoes exposed to the insecticide stimulus were more or less likely to respond than those exposed to the control stimulus. Values of relative risk were calculated for the two temperature treatments exposed to the insecticide stimulus in each replicate. In Fig 2, these data are presented in terms of log(relative risk) where values greater than zero ($> 0$) indicate mosquitoes were more likely to respond to the insecticide stimulus than to the control stimulus.

The log(relative risk) data were analyzed by linear regression models with the log(proportion responding in the insecticide stimulus treatment) as the dependent variable and log(proportion responding in the matching control stimulus treatment) as an offset term. For each dataset, the series of models started with a fully factorial model including all the interactions among the parameters; strain, temperature and dose. Subsequent models sequentially dropped one or more parameters until the null model was reached where only the intercept is estimated. These models were ranked according to Akaike's Information Criterium, AIC [51], so as to identify the 'best' model and 'near' models describing the data nearly as equally well as the 'best' model; that is models with an AIC within a value of two of the 'best' model [52,53]. As we were interested in the trend of mosquito responses with dose, rather than the response associated with any particular dose, we analyzed dose as a continuous parameter on the log-scale, log(dose). Full details of the models tested can be found in the supplementary materials.

Generalized linear models (GLM) for binomial data from the R package 'lme4' [54] were used to analyze the data from field collected mosquitoes. Two-way models with an interaction term compared the proportions of mosquitoes responding to a stimulus treatment (insecticide vs. control) and according to (i) their species, or (ii) their *kdr*-resistance genotype. Note that mosquitoes were genotyped for species and *kdr*-resistance after being behaviorally tested and it was not always possible to resolve both genotypes for each mosquito, resulting in unequal sample sizes across treatments in (i) and (ii).

## Results

### *An. gambiae* laboratory susceptible (kis) and resistant (kdrkis) strains' takeoff response after exposure to permethrin, deltamethrin and alpha-cypermethrin headspace

The relative risk ratio comparing the response to permethrin headspace relative to the response to the control headspace is represented in Fig 2A. Depending on the insecticide, the temperature and mosquito colony, the response of mosquitoes to the insecticide headspace stimulus was significantly greater than to the control stimulus, from the dose 4 or 40μg (95% confidence intervals zone not crossing the 0 axe).

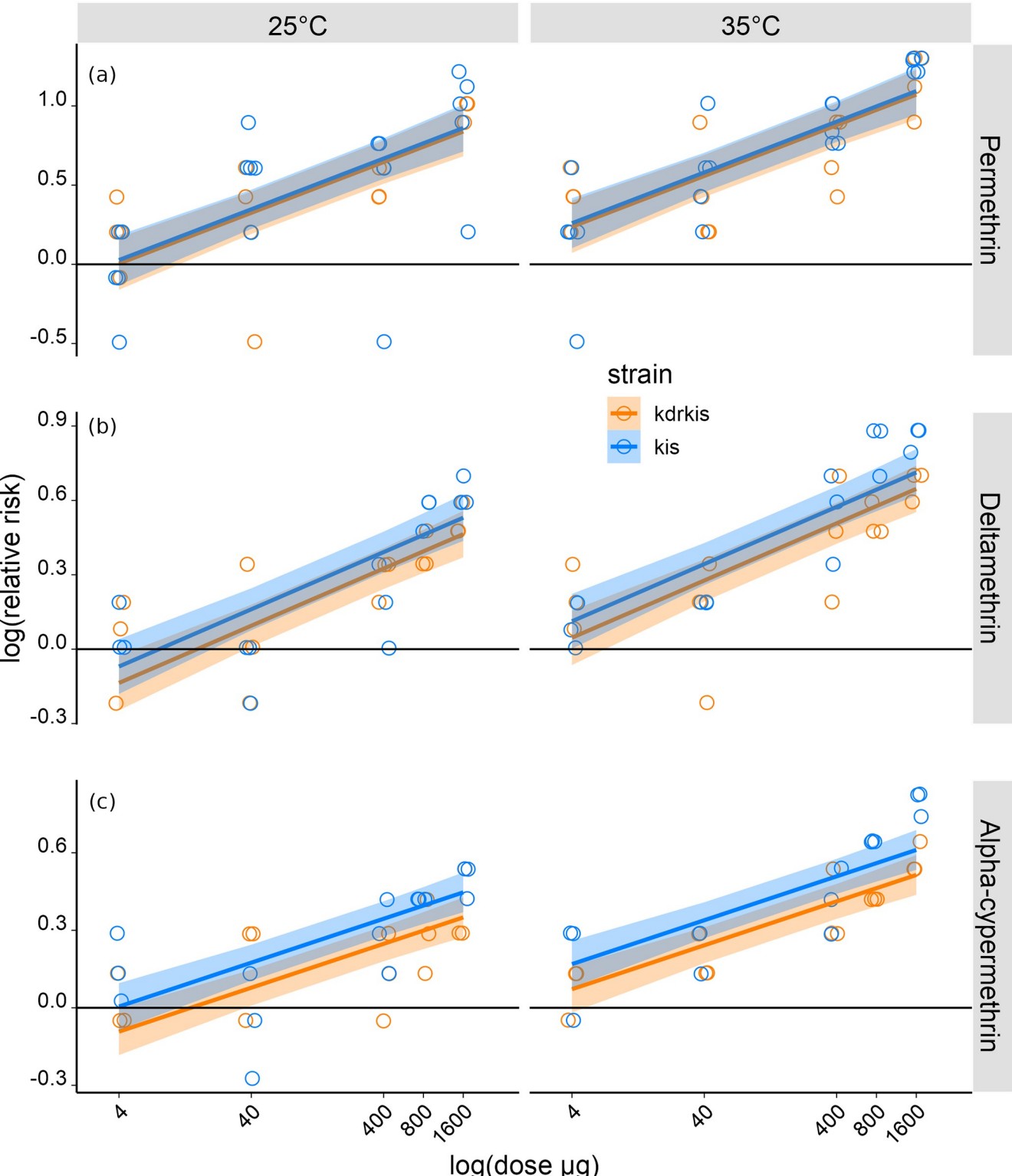

**Fig 2. Relative risk of mosquitoes taking off when exposed to three pyrethroids headspaces *vs.* their response to the headspace of the control stimulus (acetone). a)** The relative risk of *An. gambiae* kis (50 females tested per category, N = 800) and kdrkis (40 females tested per category, N = 640) taking off depending on dose of permethrin and temperature. **b)** The relative risk of *An. gambiae* kis (30 females tested per category, N = 600) and kdrkis (30 females tested per category, N = 600) taking off depending on dose of deltamethrin and temperature. **c)** The relative risk of *An. gambiae* kis (30 females tested per category, N = 600) and kdrkis (30 females tested per category, N = 600) taking off depending on dose of alpha-cypermethrin and temperature. The Y-axis is

on a log-scale. The colored bands are the 95% confidence intervals for the linear regression, when the lower interval does not cross the Y-axis at zero it indicates the response of mosquitoes to the insecticide stimulus was significantly greater than to the control stimulus.

The proportion of susceptible and resistant female mosquitoes taking off in response to the permethrin headspace increased with temperature (25°C *vs* 35°C) (P < 0.001) and dose (4; 40; 400 and 1600μg) (P < 0.001), whereas the strain of mosquitoes had no effect (P = 0.763) (Fig 2A and Table 1A).

The same experiment was performed using deltamethrin as the stimulus treatment and following the same experimental conditions except that a supplementary dose of 800μg was tested. The proportion of *An. gambiae* taking off in the presence of the deltamethrin headspace increased with the dose (P< 0.001) and temperature (P< 0.001) but did not depend on the strain (P = 0.162) (Fig 2B and Table 1B).

The same pattern was observed with alpha-cypermethrin headspace with an increased relative risk of taking off at the higher temperature (P< 0.001) and as insecticide dose increased (P< 0.001) (Fig 2C and Table 1C). In contrast to permethrin and deltamethrin, alpha-cypermethrin headspace induced different responses according to mosquito strains (P = 0.012) with the resistant mosquitoes (kdrkis) being less responsive than the susceptible mosquitoes (kis) (Table 1C).

### Field *An. gambiae sl* takeoff response to permethrin exposure

We tested the takeoff responses of *An. gambiae sl* field mosquitoes exposed to the test bottle headspace containing 1600μg of permethrin at 35°C. A total of 192 individuals were exposed to either a control stimulus or an insecticide stimulus of permethrin.

**Table 1. Analysis of variance (ANOVA) based on log(relative risk) of *An. gambiae* kis and kdrkis females taking off during exposure to the test bottle headspace at different doses of pyrethroids at 25°C or 35°C.**

a) Permethrin

|             | Df  | Sum.Sq | Mean.Sq | F.value | Pr..F   |
|-------------|-----|--------|---------|---------|---------|
| Strain      | 1   | 0.009  | 0.009   | 0.091   | 0.763   |
| Temperature | 1   | 0.976  | 0.976   | 10.099  | 0.002   |
| log(dose)   | 1   | 7.069  | 7.069   | 73.169  | < 0.001 |
| Residuals   | 68  | 6.570  | 6.097   |         |         |

b) Deltamethrin

|             | Df  | Sum.Sq | Mean.Sq | F.value | Pr..F   |
|-------------|-----|--------|---------|---------|---------|
| Strain      | 1   | 0.066  | 0.066   | 2.011   | 0.162   |
| Temperature | 1   | 0.500  | 0.500   | 15.251  | < 0.001 |
| log(dose)   | 1   | 2.866  | 2.866   | 87.491  | < 0.001 |
| Residuals   | 56  | 1.835  | 0.033   |         |         |

c) Alpha-cypermethrin

|             | Df  | Sum.Sq | Mean.Sq | F.value | Pr..F   |
|-------------|-----|--------|---------|---------|---------|
| Strain      | 1   | 0.141  | 1.141   | 6.674   | 0.012   |
| Temperature | 1   | 0.405  | 0.405   | 19.141  | < 0.001 |
| log(dose)   | 1   | 1.598  | 1.598   | 75.636  | < 0.001 |
| Residuals   | 56  | 1.183  | 0.021   |         |         |

P < 0.05 is considered as significant. Df = degrees of freedom; Sum.Sq = sums of squares; Mean.Sq = mean square; F.value = value of F distribution; Pr..F = probability value of F the statistic.

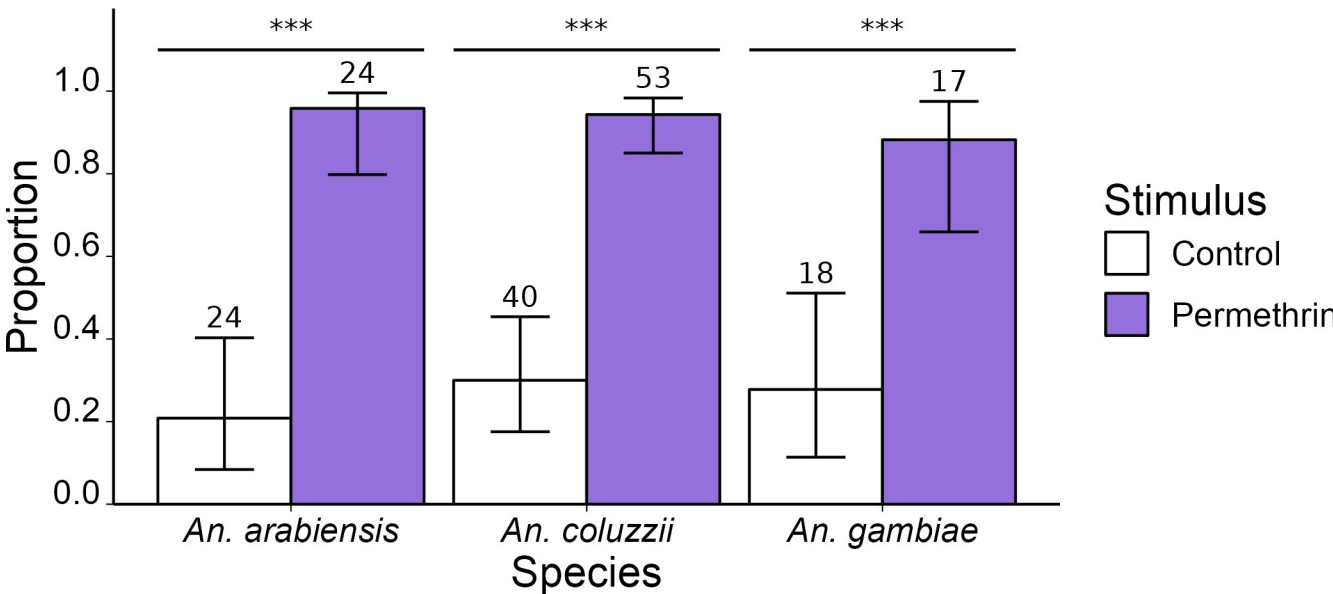

**Fig 3. Proportion of field collected mosquitoes of the Gambiae Complex taking-off in response to the test bottle headspace containing the compounds.** The numbers written in the bar plot correspond to the numbers of tested individuals. Statistically significant differences in the proportion of mosquitoes taking off are noted by asterisks ('***' = p<0.001).

Subsequent identification of the species tested found 48 *An. arabiensis*, 93 *An. coluzzii*, 35 *An. gambiae ss* (Fig 3) and 16 individuals were excluded from the analysis as their species could not be identified. Field female mosquitoes were highly responsive to the presence of insecticide, with a significant increase of females taking off in the presence of permethrin headspace for the three species (Table 2).

Genotyping of *Ace1* and *kdr* alleles evidenced the presence of Ace1(R) as well as East and West *kdr* resistance alleles (S1-S3 Tables in S1 File). Behavioral responses were analyzed only in regards to the *kdr* genotypes as they are related to the tested insecticides. For *kdr* resistant phenotypes, individuals were sequentially categorized as 'RR' if they were homozygous '*rr*' for either kdr-east or kdr-west, and then as 'RS' if one of the two loci were heterozygous '*rs*' for resistance. Individuals homozygous '*ss*' at both loci were categorized as 'SS'. Four individuals could not be categorized due to incomplete genotype data (details in S4 Table in S1 File). The behavioral response (Fig 4) was tested with regard to the three categorized *kdr* phenotypes RR (63 individuals), conferring resistance to pyrethroids, RS (38 individuals), conferring partial resistance [55,56] and SS (87 individuals) *kdr* phenotype which is not resistant to pyrethroids. The three categories showed a significant response to exposure to insecticide (Table 3).

**Table 2. Analysis of variance based on proportion of *Anopheles gambiae sl* mosquitoes responding to permethrin/control headspace.** Sum.Sq = sums of squares; Df = degrees of freedom; F.value = value of F distribution.

|  | Sum.Sq | Df | F.values | Pr(>F) |
|---|---|---|---|---|
| Species | 0.560 | 2 | 0.322 | 0.729 |
| Stimulus | 80.149 | 1 | 92.121 | 0.000 |
| Species:Stimulus | 1.107 | 2 | 0.636 | 0.541 |
| Residuals | 15.661 | 18 |  |  |

P < 0.05 is considered as significant.

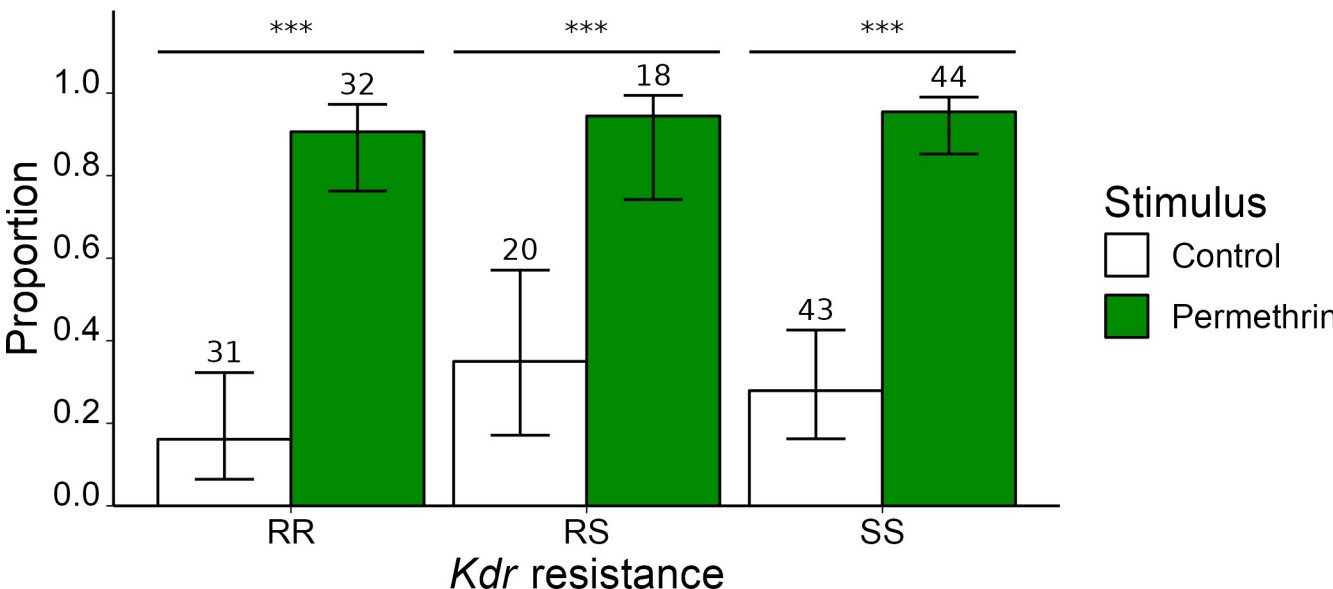

**Fig 4. Proportion of field *An. gambiae sl* flying away in response to permethrin headspace in regards to *kdr* genotypes.** The numbers written in the bar plot correspond to the numbers of tested individuals. Statistically significant differences of the proportion of mosquitoes taking off are noted by asterisks ('***' = p<0.001).

## Discussion

This study provides evidence that female malaria vector mosquitoes are able to detect the most common pyrethroid molecules currently used in vector control (i.e., permethrin, deltamethrin and alpha-cypermethrin) without direct contact, regardless of the species or their *kdr* genotype. The response to exposure from a distance to the different insecticide headspaces was significantly driven by two parameters: compound dose and temperature.

We found a positive relationship between the dose of insecticide placed in a test bottle and the risk that a mosquito would take off relative to those in a matching control treatment. Depending on the nature of the insecticide and the temperature, mosquitoes displayed a sudden locomotor response (takeoff) even when exposed at the lowest doses tested. In this experiment, the amount of insecticide carried by the airflow reaching the mosquito was not quantified, thus it is difficult to extrapolate to the real conditions found during vector control implementation. Commercially available ITNs utilize greater amounts of insecticide compared to our study. The average surface of mosquito nets available on the market is approximatively 17 m$^2$, and for instance, Olyset$^®$ contains 1000mg/m$^2$ of permethrin, Permanet$^®$ 2.0 contains 56mg/m$^2$ of deltamethrin, and 200 mg/m$^2$ of alpha-cypermethrin are used for the

**Table 3. Analysis of variance based on proportion of *Anopheles gambiae sl* mosquitoes *kdr* phenotypes responding to permethrin/control headspaces.** Sum.Sq = sums of squares; Df = degrees of freedom; F.value = value of F distribution.

| | Sum.Sq | Df | F.values | Pr(>F) |
|---|---|---|---|---|
| Kdr resistance | 2.455 | 2 | 1.446 | 0.262 |
| Stimulus | 91.182 | 1 | 107.397 | < 0.001 |
| Interaction | 0.242 | 2 | 0.142 | 0.868 |
| Residuals | 15.282 | 18 | | |

P < 0.05 is considered as significant.

Interceptor$^{\circledR}$. Yet, ITN's are designed for a slow release of the compounds by incorporating them within the fibers, and the actual quantity volatilized at a given time is expected to be much lower than the total content values. Very few studies have measured the quantity of pyrethroid released from ITNs. One of the few studies available detected 0.021μg/m$^3$ of cyfluthrin in the close environment of a treated net [57]. Another study, based on volatile collections inside deltamethrin impregnated nets reported that deltamethrin concentration vary between 0.041 μg/m$^3$ and 0.261 μg/m$^3$ in breathing zone and depending on whether the room air conditions were static or dynamic [58]. Regarding IRS, the recommended doses of insecticides spread on indoor walls are in between 20 and 30 mg/m$^2$ [59]. In this case, insecticides are not captured within the fibers, which suggests that a greater amount of insecticide will be bioavailable and potentially able to volatilize compared to those found on nets. Unfortunately, there are no studies, to our knowledge, that have quantified the insecticides present in the air after IRS implementation. The comparison between the quantity of volatilized insecticide in our experiments and the quantity present in natural conditions issued from common insecticide methods (ITN's and IRS) remains to be investigated. In any case, the goal of this study was to find evidence for the possible detection of these insecticides by malaria mosquitoes without contact, and the dose-dependent behavioral response provides it.

The observed behavioral response to insecticide headspace exposure was also temperature-dependent. Higher response rates were observed when the insecticides were exposed at a temperature of 35˚C, compared to 25˚C. Despite the expected low volatility of the tested insecticides, increasing environmental temperature may facilitate the release of a higher number of molecules, as vapor pressure increases with temperature as estimated by the Clausius-Clapeyron equation. It is worth noting that the mosquitoes in the study were placed at laboratory temperature (25±1˚C) but exposed to an airflow passing through a water bath at either 25˚C or 35˚C. This higher temperature condition could potentially influence not only the volatility of the insecticide but also the mosquito's behavior itself. Interestingly, the probability of a behavioral response to the control airflow remained consistent across different temperatures (S2 Fig in S1 File). This means that temperature of the airflow alone did not significantly impact mosquito behavior. Instead, the insecticide-dependent behavioral responses may relate to the temperature of the mosquito itself after experiencing for some time warm air coming from the airflow. This aligns with an emerging body of research highlighting the effect of temperature on mosquito olfaction [60]. The two hypothesis of an increased volatility and a higher olfactory sensitivity of the mosquito at 35˚C compared to 25˚C are not mutually exclusive and remain to be investigated.

The irritancy effect of pyrethroids after contact has been previously shown to be significantly diminished in *kdr*-resistant mosquitoes [40,61]. In this experiment, the knock-down mutation did not modify the response rates of mosquitoes exposed to permethrin. However, insecticide-susceptible mosquitoes were significantly more prone to take flight when exposed to alpha-cypermethrin than their *kdr* resistant counterparts. Similar trends were observed when exposed to deltamethrin, though the results did not reach statistical significance. These findings suggest two phenomena: i) a differential behavioral effect depending on the insecticide used, which could be the result of insecticide-specific neurotoxic effects (i.e., sodium-channel hyper activation) on mosquitoes carrying a specific *kdr* mutation (L1014F, in this study); and ii) a sensory receptor-dependent detection mechanism which is not related to the insecticide neurotoxic effect. Interestingly, recent studies in a phylogenetically distant mosquito species, *Aedes aegytpi*, have shown that repellency to pyrethrum (i.e., a natural extract containing different pyrethrins, from which pyrethroids are inspired) and to some pyrethroids is given by a dual mechanism: through a specific olfactory receptor present in the antennae of mosquitoes and also by the activation of sodium channels [62–64]. These

potential mechanisms, and their relative contributions, should also be investigated in *Anopheles* mosquitoes.

Expanding our investigation, we examined three major field-collected vector species of the Gambiae Complex in Burkina Faso: *An. gambiae*, *An. coluzzii*, and *An. arabiensis*. Remarkably, all three species exhibited takeoff responses to permethrin, irrespective of their *kdr* resistance status. This suggests that the ability to detect this particular insecticide, and potentially other pyrethroids, may be a common characteristic within this complex of species and could be prevalent among most malaria vector populations in sub-Saharan Africa. However, to confirm the occurrence of this phenomenon, further research is required in distant Gambiae populations and in other major malaria vectors, such as *An. funestus* or *An. stephensi*.

## Conclusion

This study not only provides evidence that malaria mosquitoes are able to detect insecticides without the need to get into direct contact, but unravels some of the variables that may influence such detection. The results contrast with previous experimental studies concluding the absence of distant detection by mosquitoes. The discrepancies between these results and our observations may be due to different experimental set ups and/or how insecticide was presented to the mosquitoes. Nonetheless, the results of this study highlight that dose and temperature may be critical parameters for the spatial detection of insecticides. Such experimental evidence supporting the non-contact detection of permethrin, deltamethrin, and alpha-cypermethrin by malaria vectors deserves further exploration in natural conditions. The presence of non-contact detection in several species of the Gambiae Complex opens the possibility to the potential evolution of a large range of behavioral adaptations against insecticides, which may impede or reduce the efficacy of vector control interventions and lead to the evolution of behavioral resistance.

## Supporting information

**S1 File.**
(DOCX)

## Acknowledgments

We thank Carole Ginibre and Bethsabée Scheid for mosquito rearing and Vectopôle facility management.

## Author Contributions

**Conceptualization:** Cédric Pennetier, Anna Cohuet, David Carrasco.

**Data curation:** Sassan Simplice Kambou, Philip Agnew, David Carrasco.

**Formal analysis:** Sassan Simplice Kambou, Philip Agnew, Anna Cohuet, David Carrasco.

**Funding acquisition:** Cédric Pennetier, Anna Cohuet, David Carrasco.

**Investigation:** Sassan Simplice Kambou, Adeline Valente.

**Methodology:** Sassan Simplice Kambou, Adeline Valente, Philip Agnew, Anna Cohuet, David Carrasco.

**Project administration:** Domonbabele François de Sales Hien, Rakiswendé Serge Yerbanga, Anna Cohuet, David Carrasco.

**Supervision:** Rakiswendé Serge Yerbanga, Kounbobr Roch Dabire, Cédric Pennetier, Anna Cohuet, David Carrasco.

**Validation:** Nicolas Moiroux.

**Visualization:** Sassan Simplice Kambou, David Carrasco.

**Writing – original draft:** Sassan Simplice Kambou, Anna Cohuet, David Carrasco.

**Writing – review & editing:** Sassan Simplice Kambou, Adeline Valente, Philip Agnew, Domonbabele François de Sales Hien, Rakiswendé Serge Yerbanga, Nicolas Moiroux, Kounbobr Roch Dabire, Cédric Pennetier, Anna Cohuet, David Carrasco.

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
