## [Decision Letter · Decision Letter 0]

1 Apr 2024

PONE-D-24-03507Non-contact detection of pyrethroids widely used in vector control by Anopheles mosquitoesPLOS ONE

Dear Dr. KAMBOU,

Thank you for submitting your manuscript to PLOS ONE. After careful consideration, we feel that it has merit but does not fully meet PLOS ONE’s publication criteria as it currently stands. Therefore, we invite you to submit a revised version of the manuscript that addresses the points raised during the review process.

We look forward to receiving your revised manuscript.

Kind regards,

Adekunle Akeem Bakare, Ph.D.

Academic Editor

PLOS ONE

Journal Requirements:

Whilst you may use any professional scientific editing service of your choice, PLOS has partnered with both American Journal Experts (AJE) and Editage to provide discounted services to PLOS authors. Both organizations have experience helping authors meet PLOS guidelines and can provide language editing, translation, manuscript formatting, and figure formatting to ensure your manuscript meets our submission guidelines. To take advantage of our partnership with AJE, visit the AJE website (http://aje.com/go/plos) for a 15% discount off AJE services. To take advantage of our partnership with Editage, visit the Editage website (www.editage.com) and enter referral code PLOSEDIT for a 15% discount off Editage services. If the PLOS editorial team finds any language issues in text that either AJE or Editage has edited, the service provider will re-edit the text for free.

This study was financed by the French National Research Agency – ANR (https://anr.fr), project INDEed (ANR-21-CE35-0021-01) granted to DC. The “Institut de Recherche pour le Développement” through the program ARTS provided funding for SSK doctoral studies.

4. Thank you for uploading your study's underlying data set. Unfortunately, the repository you have noted in your Data Availability statement does not qualify as an acceptable data repository according to PLOS's standards.

Reviewers' comments:

Reviewer's Responses to Questions

**Comments to the Author**

1. Is the manuscript technically sound, and do the data support the conclusions?

Reviewer #1: Yes

Reviewer #2: Yes

2. Has the statistical analysis been performed appropriately and rigorously? 

Reviewer #1: Yes

Reviewer #2: Yes

3. Have the authors made all data underlying the findings in their manuscript fully available?

Reviewer #1: Yes

Reviewer #2: Yes

4. Is the manuscript presented in an intelligible fashion and written in standard English?

Reviewer #1: Yes

Reviewer #2: Yes

5. Review Comments to the Author

Reviewer #1: Insecticide-treated nets helped to significantly reduce the malaria burden. However, their efficacy is declining due to the development and spread of insecticide resistant vectors which is causing by massive use of insecticides in vector control and selection from agricultural pesticides. There is a pressing need to improve understanding of how insecticide resistance affects the functional performance of Insecticide Treated Nets (ITNs).

The current study was conducted to evaluate the wash-resistance of PermaNet® Dual compared to WHO-prequalified pyrethroid-only (PermaNet® 2.0) and pyrethroid-chlorfenapyr (Interceptor® G2) nets by testing net pieces washed 0, 1, 3, 5, 10, 15 and 20 times in cone bioassays and tunnel tests. Tests were performed with susceptible and pyrethroid-resistant strains of Anopheles gambiae to separately assess the pyrethroid and chlorfenapyr components. Net pieces were also analysed to determine insecticide content. They show that PermaNet® Dual fulfilled WHO efficacy criteria in laboratory bioassays and showed potential to improve control of pyrethroid-resistant malaria vectors. The article is suitable for publication in Plos One Journal after addressing the minor comments below:

General comments:

It is known that cone test is not appropriate to assess efficacy of CFP-based net I am wodering why the authors strted with this method before tunnel tests. This is redondant

Abstract:

- present summary of HPLC results in the abstract

Methodology

-Describe the protocol used to wash the nets

Results

Line 323 replace contnued by continue

Line 337: which method was used?

Fig 6a how to explained that more the IG2 is washed it is inducing more knock-down?

Fig 6b how to explained the significant lower mortality of IG2 compared to P2.0?

Reviewer #2: Reports

Title: Non-contact detection of pyrethroids widely used in vector control by Anopheles mosquitoes

The author has touched on an interesting subject concerning the detection of pyrethroids without the vector being in contact. The behaviour of three strains of vectors (susceptible An. gambiae Kisumu, An. gambiae Kisumu Kdr and field An. gambiae from L1 and L2 larvae) was recorded each time they were exposed to bottles containing different insecticides from the pyrethroid class. He concludes that malaria mosquitoes are able to detect insecticides without the need for direct contact. He emphasised the influence of temperature and the different doses used in this study.

Nevertheless, the author needs to make some corrections to improve the quality of his work.

Introduction

In line 47 it’s wrote '' by decreasing both human-vector'' ref 2, this corresponds well to the reference which talks about mosquito nets. However, i would suggest that the author check whether it is pyrethrinoids that are mentioned in the report for reducing human-vector contact or mosquito nets.

Material and methods

Line 113, the author used females aged 4-8 days and not fed blood. The author should explain in one sentence why he preferred this age group of mosquitoes for the tests.

A map of the study area where the field mosquito larvae were collected would be more explicit.

On line 149, the author states that the permethrin test on laboratory mosquitoes was carried out at the IRD in France ? no equipment was needed to carry out this test in the laboratory in Burkina faso ? In other words, why weren't the tests carried out in a single laboratory because we found that the take-off rate of mosquitoes in the control group was lower with permethrin than with other insecticides, where it was higher. It would be important to clarify whether there is a particular explanation for this.

Lines 149-151 the author wrote : ''Assays involving permethrin on laboratory strains were carried out in Montpellier, France. Assays involving deltamethrin and alpha-cypermethrin on laboratory and field collected mosquitoes were carried out in Bobo Dioulasso, Burkina Faso''. However, the results showed that the field strains were only used for exposure to permethrin. If the results for the other two insecticides are not available, it would be better to correct this sentence to ensure consistency in the results.

All the additional figures are not at all legible, really small in size and do not properly facilitate understanding of the protocol.

Result

Lines 201-205 have no place in the results and have already been described in the material and methods.

Lines 272, 355 and 361 (in discussion): the author wrote ''Gambiae complex'' or ''Gambiae''; I believe the author meant to write An. gambiae complex. If so, the author should correct these sentences.

Discussion

In the discussion section, I expected the author to clarify the limitations of the control tests used given that the mosquito take-off rate was too low (less than 5%). Although it is true that the difference is significant compared to the treated bottles, other non-toxic products could activate the same reflexes in mosquitoes and it would be important to clarify this.

For clarification, was the exposure time (30 seconds) perfectly ideal? Why not shorten it by 15s or perhaps go to 1min? Could someone repeat the tests under the same conditions but slightly modify the exposure time to obtain the same result? If not, it would be more appropriate to set out the arguments in one sentence.

6. PLOS authors have the option to publish the peer review history of their article (what does this mean?). If published, this will include your full peer review and any attached files.

Reviewer #1: No

Reviewer #2: **Yes: **YOVOGAN Boulais

---

## [Author Response · Author response to Decision Letter 0]

7 Jun 2024

Dear academic editor and reviewer, 

Thank you for giving us the opportunity to submit a revised version of our manuscript entitled: “Non-contact detection of pyrethroids widely used in vector control by Anopheles mosquitoes”. We appreciate the time and effort that you and the reviewer have dedicated to providing valuable feedback on our manuscript. We are grateful to the reviewer for his insightful comments on our paper. We have been able to incorporate into this new revised version of the manuscript the suggestions provided by the reviewer. We have highlighted the changes within the manuscript. 

Below is a point-by-point response to the reviewer’s comments and concerns.

Response to Reviewer #1

Comments of reviewer 1 do not concern our paper and we notified this to Plos one which answered us through the senior editor Mr/Mrs. Miquel Vall-llosera Camps in the following words: “I am writing to follow up on your email copied below. First of all I would like to sincerely apologise for the delay you have incurred with your submission. I have discussed your concerns with the Academic Editor and we consider that, for this revision round, please only address the comments by Reviewer#2. We have suggested that Reviewer#1 should not be invited to next rounds of review and the Academic Editor might decide to invite additional reviewers if required. If you have any questions or concerns that you would like to discuss further please do let me know”.

Response to Reviewer #2

• Introduction

Reviewer comment 1: In line 47 it’s wrote '' by decreasing both human-vector'' ref 2, this corresponds well to the reference which talks about mosquito nets. However, i would suggest that the author check whether it is pyrethroids that are mentioned in the report for reducing human-vector contact or mosquito nets.

Response: Thank you for this remark. We agree with you that the cited study is about ITNs but not directly pyrethroid insecticides and the efficacy of ITN was already mentioned in the first sentence. To avoid redundancy, we removed this sentence that was not informative. 

• Material and methods

Reviewer comment 2: Line 113, the author used females aged 4-8 days and not fed blood. The author should explain in one sentence why he preferred this age group of mosquitoes for the tests.

A map of the study area where the field mosquito larvae were collected would be more explicit.

Response: Thank you for pointing this out. We now bring explanation for why we used mosquitoes aged from 4-8 in lines 123, 124 and 125 of the manuscript track-change version.

Concerning the study area where the field mosquito larvae were collected, we consider that the GPS coordinates that we provided are sufficient as all the larvae were collected from a single larval site location.

Reviewer comment 3: On line 149, the author states that the permethrin test on laboratory mosquitoes was carried out at the IRD in France? no equipment was needed to carry out this test in the laboratory in Burkina Faso? In other words, why weren't the tests carried out in a single laboratory because we found that the take-off rate of mosquitoes in the control group was lower with permethrin than with other insecticides, where it was higher. It would be important to clarify whether there is a particular explanation for this.

Response: 

Thank you for the question. The experimental protocol was initially developed in France using laboratory colonies (i.e., Kisumu and KDR-Kisumu) and permethrin, and then moved to Burkina Faso to test it in wild mosquitoes. Once there, we took the opportunity to test other insecticides with laboratory colonies. Indeed, in our study there is a confounding effect between insecticide molecule and location mainly due to the behavioral response differences between controls (i.e., permethrin vs. deltamethrin and cypermethrin), even though the experiments were performed by the same person, environmental conditions in both laboratories were standardized, and the material used and mosquito strains were exactly the same. Despite these efforts, location-dependent conditions may still exist (atmospheric pressure, water quality, etc.). However, the statistical analyses were made taking high care of these differences between sites (see statistical analysis section “Consequently, the data associated with each insecticide were analyzed separately. Line 198) and we believe that we did not over interpret our results. In the revised version, we just added “for project related reasons” from line 175 to 176 in the track change version. 

Reviewer comment 4: Lines 149-151 the author wrote: ''Assays involving permethrin on laboratory strains were carried out in Montpellier, France. Assays involving deltamethrin and alpha-cypermethrin on laboratory and field collected mosquitoes were carried out in Bobo Dioulasso, Burkina Faso''. However, the results showed that the field strains were only used for exposure to permethrin. If the results for the other two insecticides are not available, it would be better to correct this sentence to ensure consistency in the results.

Response: Thank you for this great remark. We added one sentence from line 178 to 179 in the track change version of the manuscript to ensure the consistency with the results.

Reviewer comment 5: All the additional figures are not at all legible, really small in size and do not properly facilitate understanding of the protocol.

Response: Thank you for your remark. We modified the size of the additional figures accordingly in supplementary materials track change version going from line 69 to163.

• Results

Reviewer comment 6: Lines 201-205 have no place in the results and have already been described in the material and methods. 

Response: We agree and we removed these lines (first paragraph of the results section) following your comment as it looks like a repetition.

Reviewer comment 7: Lines 272, 355 and 361 (in discussion): the author wrote ''Gambiae complex'' or ''Gambiae''; I believe the author meant to write An. gambiae complex. If so, the author should correct these sentences.

Response: Thank you for rising this point. After re-checking the current taxonomic literature, the consensual form of citing species complexes would be in our case: “Gambiae Complex”, both words written with a starting capital letter (see Harbach and Wilkerson, 2023, Zootaxa, 5303, Magnolia Press. https://doi.org/10.11646/zootaxa.5303.1.1). We have changed accordingly at lines: 25, 34, 352, 454-455 and 472 in the track change version of the manuscript.

• Discussion

Reviewer comment 8: In the discussion section, I expected the author to clarify the limitations of the control tests used given that the mosquito take-off rate was too low (less than 5%). Although it is true that the difference is significant compared to the treated bottles, other non-toxic products could activate the same reflexes in mosquitoes and it would be important to clarify this.

Response: In fact, we do not consider that the control results in the permethrin experiment are too low, but rather that the controls in deltamethrin/cypermethrin experiments are considerably high. We used acetone (the solvent used for insecticide dilutions) as control. Bottles containing the chemicals were left under the chemical hood until all the solvent was evaporated. Thus, in principle, only trace amounts of the solvent could still be present in the bottle. We have repeated the same protocol for other experiments (Kambou et al., unpublished data) using acetone and other solvents, and the control results are similar to those observed in the permethrin (France) experiment, and we have considered these results as spontaneous take-off activity with the given conditions. Unfortunately, we still do not have an explanation for why control tests in Burkina Faso ended up being higher than in France. In any case, we performed control tests for each insecticide experiment. This fact allows us to analyze and interpret the results of each insecticide test without major methodological problems. Issues could arise when trying to compare the results among insecticide tests, but this idea was already discarded at the beginning of the study simply because the different experiments could not be performed in parallel (i.e., at the same time), and temporal variables, as for instance: laboratory mosquito batches, date, atmospheric pressure, etc. could not be standardized.

We agree that other compounds, independently of their toxicity, could elicit the same take-off response. Yet, in this study we do not associate take-off response with compound toxicity, besides we consider that the experimental protocol is not suited to test this question. This experimental protocol is suited to assess a sensory detection of one stimulus (in this case chemical compounds) by mosquitoes if, and only if, there is a take-off response associated. In the absence of a behavioral response, therefore, we cannot determine that a sensory detection of the stimuli exists.

Reviewer comment 9: For clarification, was the exposure time (30 seconds) perfectly ideal? Why not shorten it by 15s or perhaps go to 1min? Could someone repeat the tests under the same conditions but slightly modify the exposure time to obtain the same result? If not, it would be more appropriate to set out the arguments in one sentence.

Response: Thank you for raising the fact that we forgot to mention the origin of our protocol. Our protocol was inspired from a previously developed assay, for testing detection of repellents (Afify et al., 2019, Current Biology) and that was able to discriminate differential detections according to molecules and conditions. We added the reference in the new version of the manuscript (line 159 in track change version- reference �49�). Nonetheless, we agree that recording the take-off time within these 30s exposure would have provided us with some temporal response information during this time period. To answer your questions: i) we did not test other exposure times, so we do not know if 30 s is the most optimal exposure time. ii) we hypothesize that 15s would give a slightly lower number of take-off responses as most of the take-offs were usually observed few seconds after the exposure. On the other hand, using 1 minute would give equal or slightly higher take-offs, in both insecticide tests and controls. iii) we are convinced that our experiments are repeatable and we encourage other researchers to validate these (or similar) questions using the same protocol. 

Responses to the journal requirements 1

Response: We carefully followed the journal template and renamed all the figures and tables in the manuscript as well as in the supplementary materials accordingly (change track versions).

Response: Our manuscript was read and corrected again by one of the co-authors, Dr. Philip Agnew research engineer at IRD who works in mosquito behavior and who is also an English native speaker. He made all the necessary in terms of language usage, spelling, and grammar formulation to make the manuscript easy to read and understandable. 

3. Thank you for stating the following financial disclosure: This study was financed by the French National Research Agency – ANR (https://anr.fr), project INDEed (ANR-21-CE35-0021-01) granted to DC. The “Institut de Recherche pour le Développement” through the program ARTS provided funding for SSK doctoral studies.

Response: We included the following statement: "The funders had no role in study design, data collection and analysis, decision to publish, or preparation of the manuscript." in the cover letter new version and in the new version of the manuscript under the “Funding” section (from line 495 to 496 in the track change version).

4. Thank you for uploading your study's underlying data set. Unfortunately, the repository you have noted in your Data Availability statement does not qualify as an acceptable data repository according to PLOS's standards.

Response: Please note the repository is listed with the 'FAIRsharing and 're3data' organisations (see links below) and is powered by 'The Dataverse Project' software developed at Harvard University.

- FAIRsharing.org - entry 'DataSuds': https://fairsharing.org/FAIRsharing.7c255a

- re3data.org - entry 'Datasuds': https://www.re3data.org/search?query=DataSuds

It is maintained by the 'Institut de Recherche pour le Developpement' (IRD), a national research institute funded by the French government. 

Data in the repository have a 'DOI' and are freely available to the public with 'CC BY' licences. Furthermore, several items already in the repository are associated with papers published in 'PLOS One' or other 'PLOS Journals' (see link below). 

 DataSuds respository, search term 'PLOS' : 

https://dataverse.ird.fr/dataverse/root?q=PLOS

Consequently, we think the repository meets the criteria suggested by PLOS and have deposited our data there. 

NOTE: The following is a confidential temporary link to the dataset deposited in 'Datasuds' that Editors and Reviewers can use to see the files: 

https://dataverse.ird.fr/privateurl.xhtml?token=0e49d898-0890-49fa-a207-a25faa33beaf

A permanent 'DOI' has been already attributed to the dataset and it will remain unchanged: DOI: https://doi.org/10.23708/IMJ1WG . This 'DOI' is given in the data availability statement in the manuscript. The DOI can only be activated once the manuscript is accepted for publication, otherwise any changes proposed by either the editor or the reviewer on this datafile would not be possible after its validation. 

We added a new section entitled Data availability in the revised manuscript (from line 498 to 501 in the track change version).

5. Please include captions for your Supporting Information files at the end of your manuscript, and update any in-text citations to match accordingly.

Response: We have taken this remark into account, as you can see from lines 452 to 453 in the track change version of the manuscript.

Response: We reviewed our reference list to ensure that it is complete and correct and we added 3 new references (references �2�, �45� and �49�) highlighted in yellow the new version of the manuscript track change version in order into account one of the reviewer 2 comments.

Responses to the journal requirements 2

1. Please remove your figures/ from within your manuscript file, leaving only the individual TIFF/EPS image files. These will be automatically included in the reviewer’s PDF

Response: We removed all the figures accordingly in the manuscript new version and leave the image files in TIFF formats 

2. We note you have provided the following data link:

https://dataverse.ird.fr/privateurl.xhtml?token=0e49d898-0890-49fa-a207-a25faa33beaf

We also note this data set is unpublished and private. Before we can proceed, please unlock this data set so that it is public and has a permanent DOI.

Response: Please note we can and will provide a permanent DOI to the files we have deposited at the DataSuds repository once our article has been accepted. Until such time, we have provided an unpublished and private link that Editors and reviewers can use to access the files in the repository. This follows the submission guidelines provided by PLOS :-

The unpublished nature of the link means we can modify the contents of the files in response to any points raised by the reviewers, e.

---

## [Editor Report · Decision Letter 1]

20 Jun 2024

Non-contact detection of pyrethroids widely used in vector control by Anopheles mosquitoes

PONE-D-24-03507R1

Dear Dr. KAMBOU,

We’re pleased to inform you that your manuscript has been judged scientifically suitable for publication and will be formally accepted for publication once it meets all outstanding technical requirements.

Kind regards,

Adekunle Akeem Bakare, Ph.D.

Academic Editor

PLOS ONE
---

## [Editor Report · Acceptance letter]

4 Jul 2024

PONE-D-24-03507R1 

PLOS ONE

Dear Dr. KAMBOU, 

I'm pleased to inform you that your manuscript has been deemed suitable for publication in PLOS ONE. Congratulations! Your manuscript is now being handed over to our production team.

Kind regards, 

on behalf of

Professor Adekunle Akeem Bakare 

Academic Editor

PLOS ONE